# Phase transformation mechanism in lithium manganese nickel oxide revealed by single-crystal hard X-ray microscopy

Saravanan Kuppan[1], Yahong Xu[2,3], Yijin Liu[2] & Guoying Chen[1]

Understanding the reaction pathway and kinetics of solid-state phase transformation is critical in designing advanced electrode materials with better performance and stability. Despite the first-order phase transition with a large lattice mismatch between the involved phases, spinel $LiMn_{1.5}Ni_{0.5}O_4$ is capable of fast rate even at large particle size, presenting an enigma yet to be understood. The present study uses advanced two-dimensional and three-dimensional nano-tomography on a series of well-formed $Li_xMn_{1.5}Ni_{0.5}O_4$ ($0 \leq x \leq 1$) crystals to visualize the mesoscale phase distribution, as a function of Li content at the sub-particle level. Inhomogeneity along with the coexistence of Li-rich and Li-poor phases are broadly observed on partially delithiated crystals, providing direct evidence for a concurrent nucleation and growth process instead of a shrinking-core or a particle-by-particle process. Superior kinetics of (100) facets at the vertices of truncated octahedral particles promote preferential delithiation, whereas the observation of strain-induced cracking suggests mechanical degradation in the material.

[1] Energy Storage and Distributed Resources Division, Lawrence Berkeley National Laboratory, Berkeley, California 94720, USA. [2] Stanford Synchrotron Radiation Lightsource, SLAC National Accelerator Laboratory, Menlo Park, California 94025, USA. [3] College of Mechanical Engineering, Donghua University, Shanghai 200051, China. Correspondence and requests for materials should be addressed to Y.L. (email: liuyijin@slac.stanford.edu) or to G.C. (email: gchen@lbl.gov).

Solid-state phase transformation is one of the key processes in the Li-ion battery technology. Conventional wisdom suggests that first-order phase transition involving large lattice misfit between the phases often leads to low-rate capability in battery cycling, yet some electrode materials, notably, nano-LiFePO$_4$ (LFP) and spinel LiMn$_{1.5}$Ni$_{0.5}$O$_4$ (LMNO), can be cycled at very high rates even in the presence of two-phase reactions with large volume changes (ca. 7 and 6.3% for LFP and LMNO, respectively)[1–3]. Much effort has been dedicated to the understanding of phase transition mechanism in LFP that led to the discovery of the non-equilibrium solid–solution pathway that bypasses the nucleation and growth process in nano-sized LFP[4–7]. Despite its technological significance as a promising cathode materials for high-energy Li-ion batteries, very few studies focused on detailed phase transformation in LMNO[8]. High-rate capability can be achieved even on micron-sized LMNO, but the relationships among kinetics, strain and phase boundary movement are currently unknown.

Structure wise, LMNO has a robust cubic close-packed spinel lattice with edge-shared MO$_6$ octahedra (M = Mn and Ni). The large difference in ionic radii of Mn$^{4+}$ and Ni$^{2+}$ promotes cation ordering in the 16$d$ octahedral sites, but the degree of ordering strongly depends on the synthesis routes and conditions[9–11]. During delithiation, LMNO converts to Mn$_{1.5}$Ni$_{0.5}$O$_4$ (MNO) by two two-phase reactions $\sim 4.7$ V versus Li/Li$^+$: LMNO (Ni$^{2+}$)$\rightarrow$Li$_{0.5}$Mn$_{1.5}$Ni$_{0.5}$O$_4$ (L$_{0.5}$MNO, Ni$^{3+}$) and L$_{0.5}$MNO$\rightarrow$ MNO(Ni$^{4+}$)[12,13]. Although the cubic framework remains unchanged among the three phases, the lattice parameter decreases from 8.17 Å in LMNO to 8.09 Å in L$_{0.5}$MNO and then 8.00 Å in MNO, leading to a net volume change of 3% (LMNO to L$_{0.5}$MNO), 3.3 % (L$_{0.5}$MNO to MNO) and 6.3% (LMNO to MNO). In addition, depending on the structural ordering, solid–solution regimes with various sizes can exist at both high and low-state of charge (SOC) ends[14]. Recently, Takahashi et al.[15] used temperature-controlled in operando X-ray absorption spectroscopy (XAS) on LMNO and revealed a total phase transformation activation energy of 75 kJ mol$^{-1}$ (29 and 46 kJ mol$^{-1}$ for LMNO to L$_{0.5}$MNO and L$_{0.5}$MNO to MNO, respectively), which is surprisingly large compared to the other well-known intercalation-based electrode materials[15].

The large lattice mismatch between the phases and the associated elastic deformation likely lead to strain at the phase boundaries[16] and induce mechanical fracture and failure, which is one of the known mechanisms for energy and power loss, and poor cycle life in batteries[6,7]. High spatial- and chemical-resolution imaging of phase boundary propagation at the particle level is important, as it provides direct evidence on phase transformation mechanism and the associated mechanical damage/fracture, which in turn guides us on the design of particle morphology and other properties to achieve improved electrode performance. When combined with electron diffraction and/or electron energy loss spectroscopy, high-resolution transmission electron microscopy has traditionally been used to probe structural and chemical changes at nanoscale[17]. However, these studies are handicapped by the stringent sample requirements (thickness of <100 nm, for example), high-vacuum environment and the limitation in sample size. In recent years, hard X-ray full-field transition microscopy imaging combined with X-ray absorption near-edge structure (FF-TXM-XANES) was introduced as a revolutionarily new approach for visualizing electrochemically driven solid-state phase transformation[18–21]. The brightness and the energy tunability of synchrotron-based hard X-ray enable nanoscale spatial resolution at $\sim 30$ nm along with high chemical and elemental sensitivities in a large field-of-view (FOV; $30 \times 30 \mu$m). Newer development in FF-TXM-XANES capability has enabled spectroscopic data acquisition rate several

orders of magnitude faster than what was possible a few years ago[20,22,23]. Research on scientific big data mining methods aiming for more effective analysis and better interpretation of the X-ray spectro-microscopic data has also been actively pursued[24–27]. Recent studies successfully utilized these advances in FF-TXM-XANES, and revealed the nucleation and growth process of new phases as a function of Li content in conventional secondary cathode particles, including LFP[28–31], LiMn$_2$O$_4$ (ref. 32), FeF$_3$ (ref. 19), LiNi$_x$Mn$_y$Co$_{1-x-y}$O$_2$ (refs 33–36) and Li$_2$Ru$_{0.5}$Mn$_{0.5}$O$_3$ (ref. 37).

In this work, we prepare a series of micron-sized (with an average size of 3 μm) and octahedron-shaped Li$_x$Mn$_{1.5}$Ni$_{0.5}$O$_4$ (L$_x$MNO) crystals with dominant (111) family surface facets and employ FF-TXM-XANES to visualize the mesoscale phase distribution at a single-crystallite level for the first time. Through quantification combining X-ray diffraction and synchrotron spectroscopy, phase-pure XANES spectra for all three principle chemical species in LMNO, namely, LMNO, L$_{0.5}$MNO and MNO, are obtained. The phase-pure intermediate phase, L$_{0.5}$MNO, has not been synthesized experimentally and here we report on its standard X-ray absorption spectrum. The use of a solution-based chemical oxidation process allows us to ensure a homogeneous delithiation conditions for all the particles which eliminates the effect of electrical contact due to poor contact between particles and the current collector typically seen in electrochemical charging. The intrinsic phase transformation behaviour in LMNO are then probed and revealed. It is found that before propagating into the inner part of the crystal, delithiation preferentially initiates at the truncated (100) surface planes where Li$^+$ transport is more favourable. Extensive particle fracturing is only observed on fully delithiated large MNO particles. Critical insights for designing better performing and more stable LMNO cathodes are also obtained.

## Results

**Synthesis and bulk characterization of crystal samples.** Pristine LMNO crystals with a cubic spinel phase and structurally ordered in a $P4_332$ space group were synthesized by the molten-salt method[3]. Previous electrochemical studies showed that the sample delivered $\sim 140$ mAh g$^{-1}$ at C/5 rate and experienced an average per-cycle loss of $\sim 0.05\%$ and a coulombic efficiency of 99% in the first 170 cycles[38]. Supplementary Fig. 1a shows the X-ray diffraction patterns of a series of L$_x$MNO ($x = 0.90$, 0.82, 0.71, 0.51, 0.40, 0.25, 0.11, 0.06 and 0) crystals prepared by chemical oxidation of LMNO with varying amounts of a 0.1 M solution of nitronium tetrafluoroborate (NO$_2$BF$_4$) in acetonitrile, a procedure previously reported[39]. The evolution among the three cubic phases with the same crystal structure, but different lattice parameters is clearly evidenced by the distinct sets of reflections well separated from each other. Careful full-pattern Rietveld refinement produced lattice parameters of 8.1687(2), 8.0910(6) and 8.0005(3) Å for the LMNO, L$_{0.5}$MNO and MNO phases, respectively, consistent with the reported values in the literature[13]. The obtained weight fraction of the phases in each sample was plotted as a function of the lithium content ($x$) determined by inductively coupled plasma analysis and shown in Supplementary Fig. 1b. At high SOC ($0.71 < x < 1$), L$_x$MNO samples are composed of LMNO and L$_{0.5}$MNO only and the phase ratio between them was 69:31 at $x = 0.82$. MNO appeared on further Li removal and coexisted with the other two phases in the samples with an intermediate SOC ($0.25 < x \leq 0.71$), although only 3% exist in the sample with $x = 0.71$. At low Li content ($x \leq 0.25$), the samples were composed of L$_{0.5}$MNO and MNO only and the phase ratio between them was 59:41 at $x = 0.25$. Due to the large step size and the coarse nature of the chemical

delithiation approach, it is unclear whether a solid–solution region existed in the high Li content region of $0.9 < x < 1$, before the domination of the biphasic transitions.

Figure 1a shows the Ni K-edge XANES spectra of the $Li_xMn_{1.5}Ni_{0.5}O_4$ series collected at beamline 4-1 at the Stanford Synchrotron Radiation Lightsource (SSRL). With decreasing $x$, there is a monotonous edge shift to higher energy, consistent with a continuous increase in average Ni oxidation state, as it transitioned from the divalent state in LMNO to tetravalent state in MNO. The linear relationship between Ni edge energy (from 8342.6 to 8346.5 eV) and SOC of the samples is clearly shown in Fig. 1b, suggesting that Ni oxidation state may be used as a proxy for sample SOC determination. Note that in this work, XANES edge position is defined by the interception of spectrum at 1/2 of the normalized intensity. Two isosbestic points, indicative of a three-component system, were observed at 8353.8 and 8355 eV, consistent with the presence of LMNO, $L_{0.5}MNO$ and MNO principle chemical species in the samples[35]. While LMNO and MNO end members can be prepared by the chemical delithiation or an electrochemical method, $L_{0.5}MNO$ coexists with one or two other phases throughout the entire Li content range, as shown in

the phase diagram in Supplementary Fig. 1b. Experimentally, synthesizing and isolating phase-pure $L_{0.5}MNO$ was found impractical. As the weight fractions of the three phases in the samples at different SOCs can be obtained from our X-ray diffraction pattern refinement, the spectrum of the phase-pure intermediate principle chemical species, $L_{0.5}MNO$, was therefore extrapolated numerically via mathematical manipulation[40]. Specifically, the spectra of phase-pure LMNO and MNO ($S_1$ and $S_0$) were acquired directly. All of the experimentally measured spectra of partially dilithiated $L_xMNO$ ($S_x$, $x = 0.06$, 0.25, 0.51, 0.71, 0.82, 0.9) were composed of varying contributions from the three primary phases (LMNO, $L_{0.5}MNO$ and MNO), with the relative weight fraction of each phase at a given $x$ value ($R_x^1$, $R_x^{0.5}$ and $R_x^0$) obtainable from the Rietveld refinement of the full X-ray diffraction patterns. This gives us six redundant linear equations that can be expressed as $S_x = (S_0 \times R_x^0) + (S_{0.5} \times R_x^{0.5}) + (S_1 \times R_x^1)$. The only unknown parameter in these equations is $S_{0.5}$, which is the signature spectrum of the phase-pure $L_{0.5}MNO$. $S_{0.5}$ was then solved by minimizing the deviation among the results derived from multiple independent experimental measurements, while satisfying the constraints set by the equation

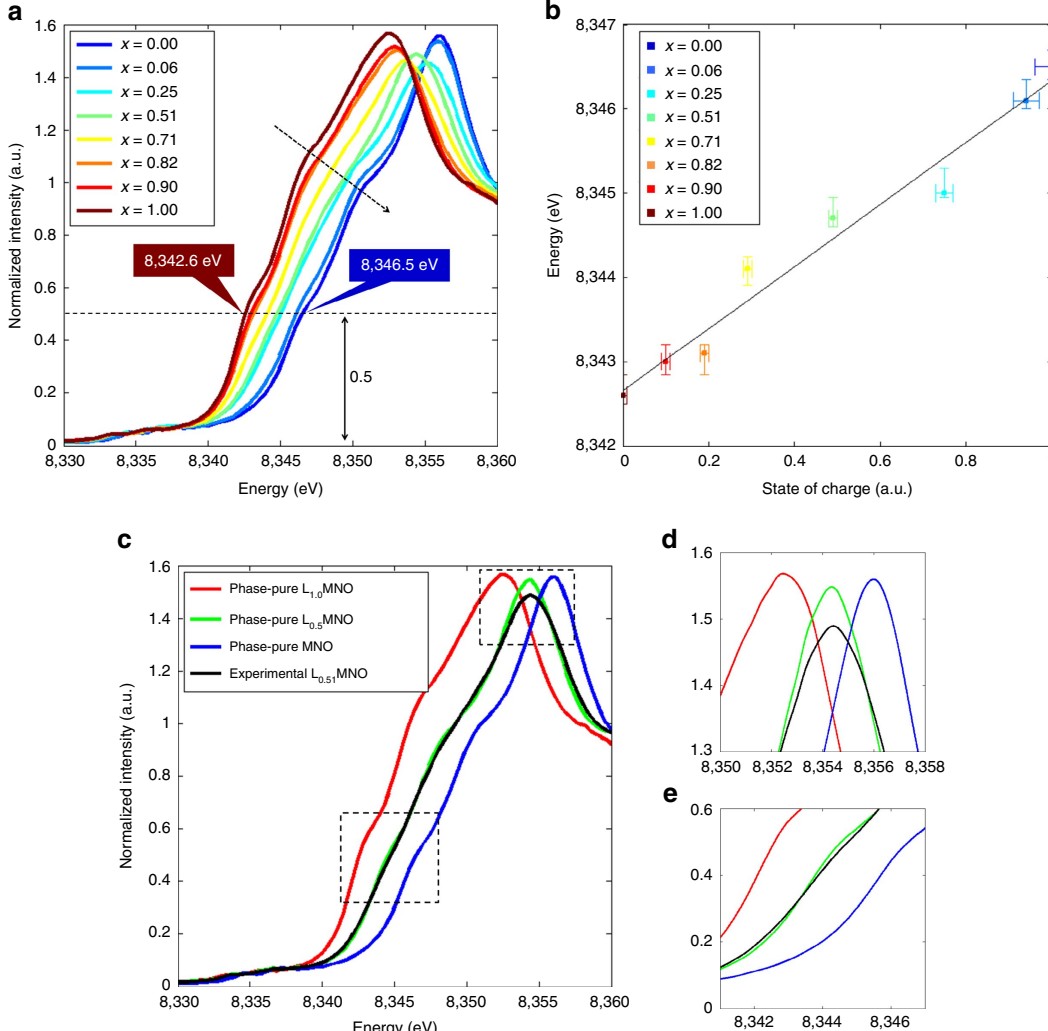

**Figure 1 | XANES measurement of the crystal samples.** (**a**) Experimentally measured Ni K-edge XANES spectra of chemically delithiated $Li_xMn_{1.5}Ni_{0.5}O_4$ ($Li_xMNO$) crystals, (**b**) Ni edge energy as a function of the state of charge in $Li_xMNO$ samples, (**c**) Ni K-edge XANES spectra of phase-pure $LiMn_{1.5}Ni_{0.5}O_4$ (red), $Li_{0.5}Mn_{1.5}Ni_{0.5}O_4$ (green) and $Mn_{1.5}Ni_{0.5}O_4$ (blue) phases. The experimentally measured spectrum of $Li_{0.51}Mn_{1.5}Ni_{0.5}O_4$ (black) is also shown for comparison, **d,e** enlarged views of the two highlighted regions in **c**. Horizontal error bars in **b** were obtained by quantifying the SOC multiple times. Vertical error bars in **b** were obtained by evaluating the energy resolution of the beamline used for the XANES measurements.

described above. The calculation was implemented using Matlab's optimization tool box.

Figure 1c shows the XANES spectra of the three phase-pure principle chemical species in the LMNO system, which represents the first report on the XANES spectra of the phase-pure $L_{0.5}$MNO intermediate phase. For comparison, experimentally measured spectrum from the $L_{0.51}$MNO sample, which consists of a mixture of LMNO, $L_{0.5}$MNO and MNO phases at 19.3, 67.6, and 13.1%, respectively, is also shown in Fig. 1c. Two enlarged regions shown in Fig. 1d,e clearly show that the numerically generated $L_{0.5}$MNO single-phase spectrum has more pronounced near-edge features, especially at the white line (Fig. 1e), while these features appear mostly smoothed out in the experimentally measured spectrum. The near-edge features in the XANES spectrum is known to be sensitive to the spatial arrangement of the atoms near the targeted centre atom (Ni atom in this case), extensive analysis is therefore currently underway to fully understand the structural arrangement in the $L_{0.5}$MNO intermediate phase. Nonetheless, the observed difference between the experimentally measured and the numerically retrieved spectra of $L_{0.5}$MNO highlights the importance of obtaining the reference spectrum for each principle chemical species. Figure 1c also shows that the edge shift from LMNO to $L_{0.5}$MNO and $L_{0.5}$MNO to MNO was $\sim 2$ eV each, consistent with an average Ni oxidation state of $2+$, $3+$ and $4+$, respectively.

**Analysis of particle-level phase distribution**. To investigate the chemical homogeneity in $L_x$MNO at (sub) particle level, we conducted a series of FF-TXM studies at the beamline 6-2c at SSRL. The details of the experimental set-up and the data processing software, known as TXM-Wizard, can be found in the literature[26]. The transmission images acquired at the pre-edge region of the Ni K-edge (at 8,335 eV, where the absorption coefficients are independent of the oxidation state) are capable of providing morphological information of the micron-sized crystals. In addition, chemical sensitivity can be introduced by performing TXM imaging at different X-ray energies. Series of FF-TXM images of the $L_x$MNO crystals were acquired as the X-ray energy was tuned across the Ni K-edge. After proper imaging normalization and registration, the spectra over each single pixel at $\sim 30$ nm were constructed and subsequently fitted using a linear combination of the standard spectra of the three phase-pure principle chemical species (LMNO, $L_{0.5}$MNO and MNO) shown in Fig. 1c. Typical FOV of the FF-TXM images of $Li_{0.82}$MNO, $Li_{0.71}$MNO and $Li_{0.51}$MNO crystal samples are shown in Supplementary Fig. 2. Images were acquired in the transmission mode and therefore the contrast is directly proportional to the mass thickness of the crystals. Dark features on the images are generally indicative of microstructural fractures or cracks within the crystals. A large number of crystals (which are numerically labelled in Supplementary Fig. 2) were analysed to ensure the statistics accuracy of our study, but for simplicity, only the representative crystals in each sample are presented here for detailed discussion.

Figure 2a–d shows the chemical phase distribution on four different particles with a SOC of $x = 0.82$, 0.71, 0.51 and 0.25, respectively. The two-dimensional (2D) chemical maps were generated by linear combination fitting of the standard XANES spectra of the three principle chemical species. The chemical maps are colour coded according to the colour legend shown in Fig. 2a, where red, green and blue represent LMNO, $L_{0.5}$MNO and MNO phases, respectively. In the mildly delithiated samples (such as $L_{0.82}$MNO), both LMNO and $L_{0.5}$MNO phases were observed in the chemical map (Fig. 2a), which is consistent with our observation in the bulk X-ray diffraction. The detection of the

two phases at sub-particle level reveals the heterogeneous nature of the delithiation process even though homogenous oxidation conditions were used in our study. Further delithiation leads to the coexistence of all three phases in both $L_{0.71}$MNO (Fig. 2b) and $L_{0.51}$MNO (Fig. 2c) crystal samples. Lithium-poor domains appeared along with the lithium-rich domains in all four particles, as shown in the phase distribution maps in Supplementary Fig. 3, which provides clear evidence for the concurrent nature of the process. These results invalidate the notion that LMNO delithiation occurs through the classic shrinking-core process, where the lithiated phase is expected to be at the core/centre of the particle and surrounded by a delithiated layer on the particle surface. The more extensively delithiated particles ($Li_{0.25}$MNO, Fig. 2d) only consist of $L_{0.5}$MNO and MNO phases without detectable LMNO, consistent with the bulk X-ray diffraction result.

To further illustrate the chemical heterogeneity on these particles, spectra collected over selected small region of interest (average of six pixels by six pixels) along the arrows highlighted on the images in the corresponding insets are shown in Fig. 2e–h. The monotonous Ni absorption edge shift to the lower energy corresponds to a reduction in Ni oxidation state. It concurs with an increase in the weight ratio of the lithiated phase(s) and, subsequently, a decrease in SOC along the direction of the arrows. Figure 2i–l shows the weight percentage of the three principle chemical species, as well as the Ni edge energy change as a function of the position along the arrows highlighted in Fig. 2e–h. The correlation between Ni oxidation state and SOC is evident, further confirming the feasibility of using Ni oxidation state mapping for visualizing SOC distribution at the sub-particle level.

With the 2D transmission geometry, the chemical maps shown in Fig. 2 integrate the signal along the beam path and therefore do not contain the information in the third spatial dimension (that is, the depth). To understand the phase distribution in real three-dimensional (3D) space and subsequently the delithiation pathway in LMNO, we performed the well-established 3D XANES study[17,26] on selected partially delithiated particles. The results on one of the $L_{0.51}$MNO particles are shown in Fig. 3, where the data set is essentially four-dimensional with the X-ray edge energy presented in a colour coding on top of the three spatial dimensions ($x$, $y$ and $z$). The shape of the particle is reconstructed by the tomography measurement of the data acquired at single X-ray energy at 8,960 eV and presented here using the transparent surface rendering in all the panels in Fig. 3. The persistent edge energy, which is associated with the Ni oxidation state and an indicator of the SOC, further reveals the heterogeneity in the partially delithiated particles. Figure 3b–d shows the distribution of edge energy over the diagonal slices through the centre of the particle, which clearly show that the six corner vertices (shown in Fig. 3a) are at different SOCs even though the entire particle was submerged in the same oxidation medium during chemical delithiation. The two blue corner vertices are more oxidized, indicating possible preferential delithiation at these two locations. The two green vertices are only partially oxidized, while the two red ones remain at their low oxidation state. In addition to the slicing views, the propagation of the reaction front from the blue vertices to the red vertices is visualized by the 3D distribution mapping of the Ni oxidation state in Fig. 3e–j, where the segmentation of 3D-sub-volume associated with the different edge energy within the particle is shown. Figure 3e is the overall view of the segmentation results, while Fig. 3g–j are separate views of the 3D-sub-volumes at increasing edge energy from $\sim 8,342.5$ eV ($Ni^{2+}$) to $\sim 8,346.5$ eV ($Ni^{4+}$). Figure 3f,h,j therefore represent the regions with edge energies that are associated with one of the three principle chemical phases (LMNO, $L_{0.5}$MNO and MNO), while Fig. 3g,i

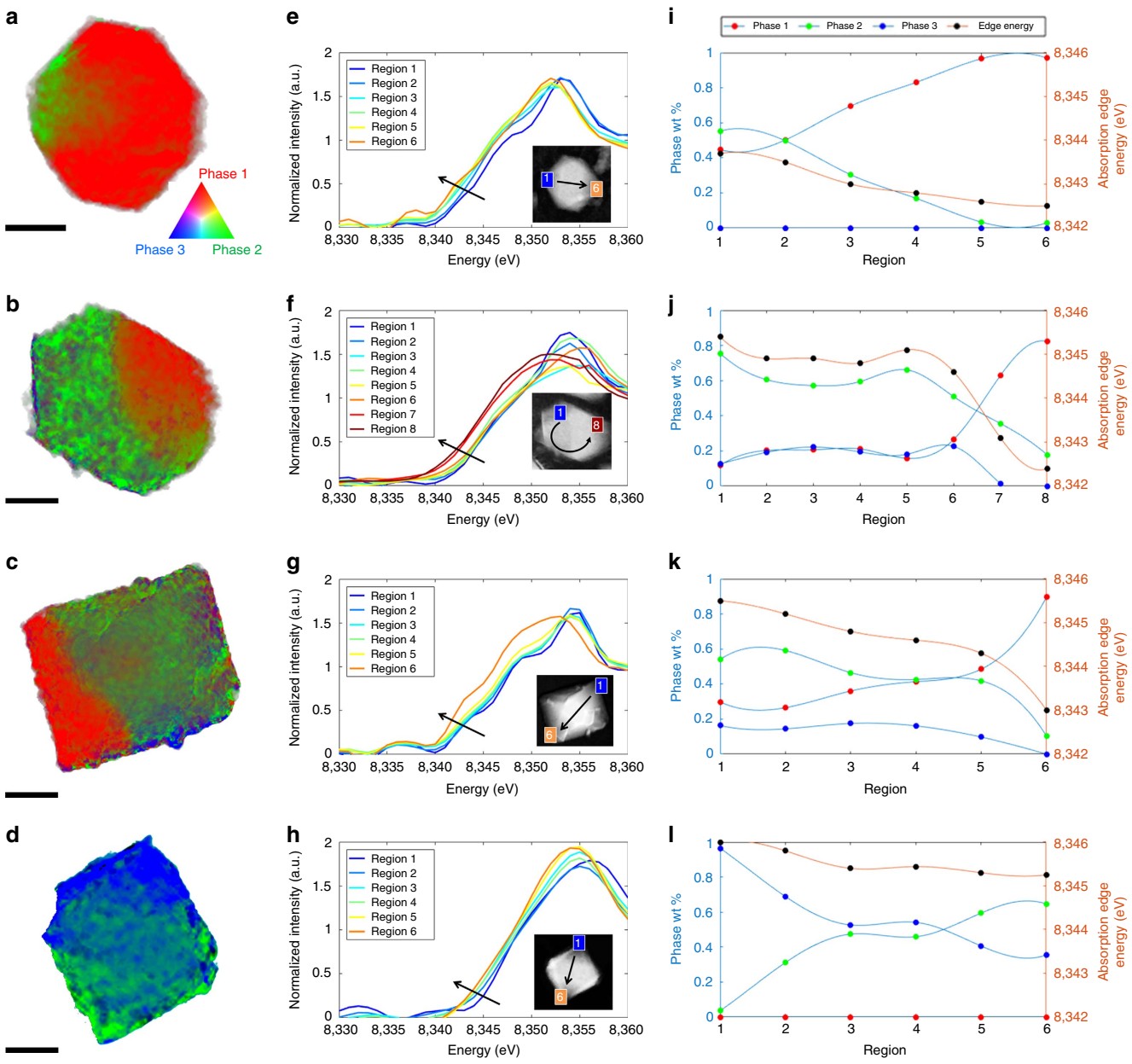

**Figure 2 | Two-dimensional chemical mapping of $Li_xMn_{1.5}Ni_{0.5}O_4$ crystals.** (**a**) $x = 0.82$, (**b**) $x = 0.71$, (**c**) $x = 0.51$ and (**d**) $x = 0.25$. (**e–h**) the variation of local XANES spectrum across each particle. (**i–l**) the relative concentration of the three chemical phases, as well as the variation in the local oxidation state of Ni within the same particle. Scale bar, 0.5 μm.

represent the regions with mixed phases. Note that the observed 3D-sub-volume with mixed phases could also be caused by the fine domains that are beyond the spatial resolution limit of this method at $\sim 30 \times 30 \times 30\,nm^3$. Nonetheless, the trend that the reaction front migrates from the blue corner vertices, where delithiation preferably initiated, to the red region of the particle is clearly observed. The domains of different SOCs are intermixed instead of segregated within the particle, further confirming our observation in the 2D chemical maps in Fig. 2.

**Phase transformation mechanism.** It is well established that LMNO, $L_{0.5}$MNO and MNO phases have similar chemical potentials and they coexist at a flat voltage of 4.7 V versus $Li/Li^+$ upon Li extraction[35]. The lack of thermodynamic driving force in redistributing the phases after delithiation suggests that the

particle-level chemical distribution maps are fossil evidence for the dynamic phase transition process in LMNO. Here we examine the possible origin of the heterogeneity occurring during the phase transformation. Structurally ordered LMNO single crystals synthesized by the molten-salt method primarily consist of [111] family of planes that are the most stable surface facets in the spinel structure based on the density functional theory (DFT) calculations[41,42]. Slight truncation at some of the corner vertices was also observed in our crystals, leading to an appearance of somewhat truncated octahedron crystals and the exposure of (100) surface planes, as shown in the scanning electron microscopy images in Fig. 4a (right). In contrast to the (111) plane, the (100) plane in the ordered LMNO has less dense arrangement of the transition metal atoms, as shown in the comparison in Fig. 4b left and right. This leads to higher-surface energy and more vulnerability for Mn dissolution on (100)

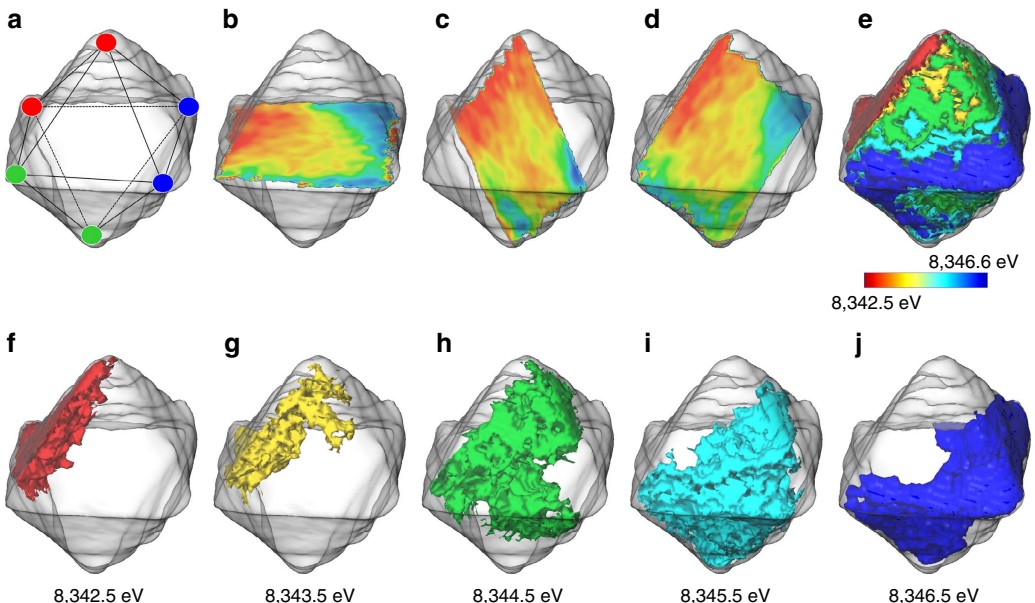

**Figure 3 | Three-dimensional map of Ni oxidation state at sub-particle scale.** The shape of the particle is presented as the transparent grey surface with the internal oxidation state heterogeneity illustrated using the diagonal slices (**a**–**d**) surfaces of the 3D Ni oxidation state map (**e**–**j**). All the panels are colour coded in order to show the state of charge heterogeneity at the sub-particle level in a quantitative manner.

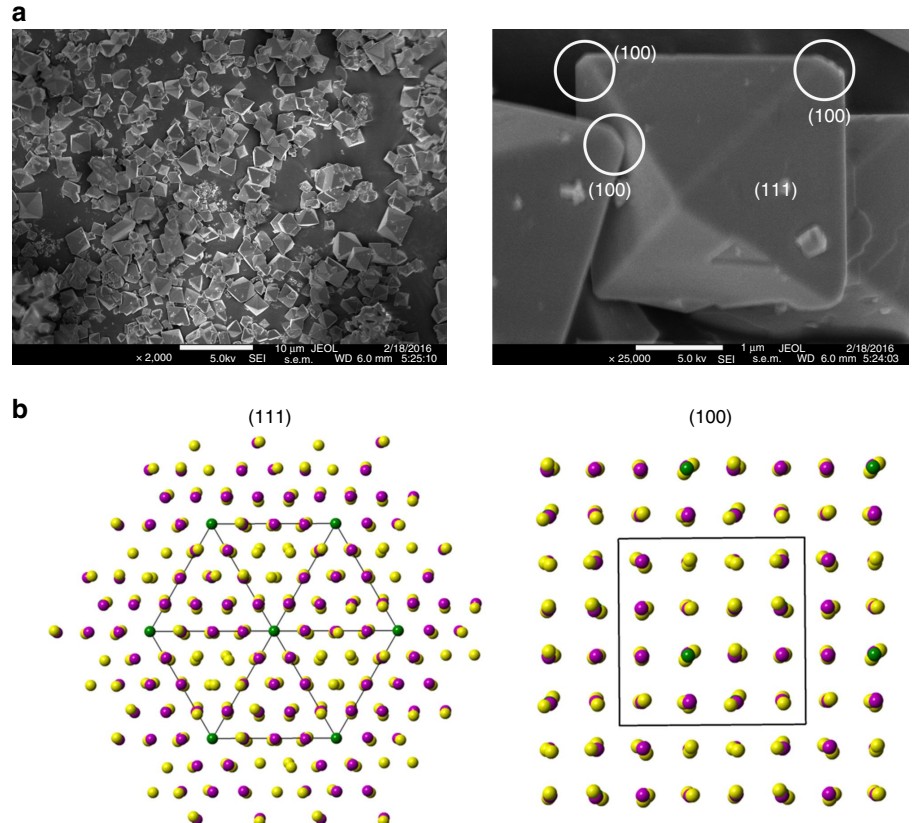

**Figure 4 | Surface facets of LiMn$_{1.5}$Ni$_{0.5}$O$_4$ crystals. (a)** Scanning electron microscopy images showing the truncation at some corners of the crystals and **(b)** atomic models of the (111) and (100) planes in the ordered spinel with a $P4_332$ space group: green balls (Ni cations), magenta balls (Mn cations) and yellow balls (O anions). Scale bars, 10 and 1 μm in **a**, left and right, respectively.

planes[43], but in the meantime, they are also expected to have more facile Li$^+$ transport pathways. The superior kinetics of (100) planes as compared with that of the (111) planes was experimentally confirmed on both spinel LiMn$_2$O$_4$ and LMNO

cathode materials[19,44,45]. We believe that this kinetic advantage leads to preferential nucleation of the delithiated phase at the vertices where (100) facets are located. Once the nucleation is initiated, the growth of the new phase then dominates the

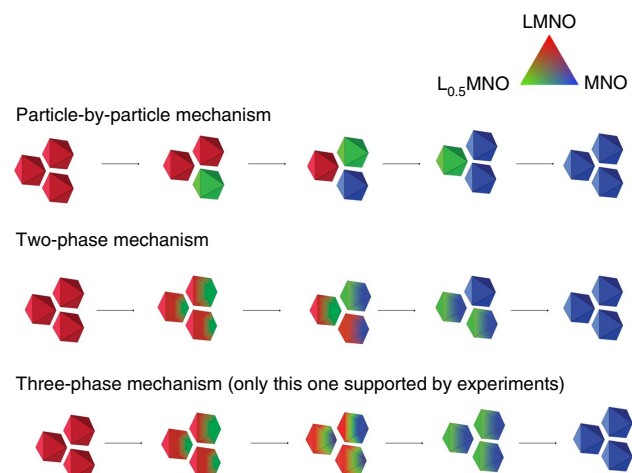

**Figure 5 | Schematics of possible phase transformation mechanisms in LiMn$_{1.5}$Ni$_{0.5}$O$_4$.**

transformation and the reaction quickly propagates into the bulk of the crystal. This anisotropic nucleation-growth model, as opposed to the classic shrinking-core, is clearly supported by the chemical phase distribution maps in both Figs 2 and 3.

On the basis of our observations through the 2D and 3D FF-TXM-XANES investigation, a schematic illustration of the possible nucleation-growth pathways in LMNO particles is shown in Fig. 5. Truncation in our crystals plays a crucial role in kinetics as the nucleation of the delithiated phase at the truncated vertices initiates the phase transformation. As the growth of the new phase inside the crystal is considerably faster than further nucleation at the (111) facets, the result is inhomogeneous, intra-particle coexistence of multiple phases. Individual particles that are single phased, either fully delithiated or fully lithiated, was not observed in our partially delithiated samples, regardless of the particle size. This invalidates the particle-by-particle phase transition mechanism shown in Fig. 5 (top). Further, at the intermediate SOCs, all three phases were detected by the bulk measurements and the X-ray microscopic imaging at the single-particle level. This observation clearly invalidates the two-phase mechanism shown in Fig. 5 (middle). The study, therefore, unanimously supports a three-phase concurrent phase transfor-mation mechanism in LMNO, as shown in Fig. 5 (bottom).

We would like to point out that the chemical delithiation condition used in this study is similar to that of constant-voltage electrochemical charging under an overpotential. The standard oxidation potential of the $NO_2^+/NO_2$ redox couple is ~5.1 V versus Li$^+$/Li, which is ~400 mV higher than that of the plateau potential of the LMNO cathode at ~4.7 V. The presence of such a large overpotential is likely to promote a high Li flux during the extraction, which may lead to significant variations in local concentration and contribute to the particle-level heterogeneity observed in this study. Furthermore, the delithiation process under these conditions is likely limited by the bulk diffusion of Li$^+$ that may lead to the initiation of the second-phase transformation (L$_{0.5}$MNO to MNO) at the outer surface before the first-phase transformation (LMNO to L$_{0.5}$MNO) in the entire particle can complete. This further contributes to the particle-level heterogeneity observed in this study. It is possible that under low constant-current charging (small C-rate) with only a small overpotential, other phase transformation mechanism could dominate or coexist. Systematic studies using different experi-mental conditions, especially the use of *operando* techniques such as those recently demonstrated on LFP[30] should provide further information on phase transformation mechanism(s)

under electrochemical charge/discharge conditions. One caution is that in electrochemical charge/discharge, Li extraction is greatly influenced by ionic and electronic connectivity that introduces additional variations contributing to the observed delithiation mechanism at the particle-level. Whereas in chemical delithiation, all LMNO particles are immersed in the oxidizing NO$_2$BF$_4$ solution with the same concentration and therefore, delithiation uniformity at both particle level and bulk level can be achieved.

**Mechanical strain and stress release.** Coexistence of multiple phases on a single particle was previously reported in the LFP system[4,24,34], lattice mismatch between the phases results in mechanical strain at the phase boundaries, which is often released by dislocations or fractures on the particle. Extent of delithiation and strain-induced fracturing were found to be particle size dependent in LFP, with less fracturing at high level of delithiation observed on the smaller particles[46]. A recent study using the *in situ* coherent X-ray diffractive imaging technique showed that the strain energy is relatively low at the early stages of LMNO delithiation, but it increases >10-fold during the topotactic phase transformation process[16,47]. Striking inhomogeneity in the strain distribution throughout the particle was also reported. Further evidence was provided by *in situ* multi-beam optical stress sensor on LMNO thin-film electrodes, which showed that the stress in LMNO increases with lithium removal and it can reach up to 126 MPa at fully delithiated state (MNO)[48]. Strain survey on Li$_x$MNO crystals was not feasible with FF-TXM as the contrast mechanism of the technique is not sensitive to the local crystal structure distortion. However, morphology survey on the particles, such as the formation of cracks, can be carried out especially if the microscope is operated at lower X-ray energy than that of the Ni K-edge. This is because better absorption contrast and higher spatial resolution are achievable at lower energy. Figure 6a shows the three-dimensional surface rendering of two selected fully delithiated MNO crystals (4 and 2 µm in size) prepared by chemical oxidation. The formation of irregularly shaped cracks on the particle surface is clearly shown. Further evidence on delithiation-induced morphological changes was provided by scanning electron microscopy studies, as shown by the images obtained on pristine LMNO and chemically delithiated MNO in Fig. 6b,c, respectively. No changes were observed on Li$_x$MNO crystals at low SOC ($0.51 < x < 1$) and only minor cracks developed on a few random particles in the samples with $0.25 < x < 0.51$. Severe particle fracturing on a large number of particles was only observed on fully delithiated MNO (Fig. 6b), where a large volume change of 6.3% from the pristine LMNO to MNO is present. In general, large-sized cracks surrounded by a number of small cracks were observed over the entire (111) surface facets, but systematic trend in the generation or propagation of the cracks was not found. The observation of fractures in our large MNO crystals with 3 µm size confirms the accumulation and relaxation of strain due to the substantial volume change and phase boundary movement on the transformation from LMNO to MNO[12,13]. Fracturing, however, was not observed on smaller crystals (<1 µm) even at fully delithiated state (MNO), suggesting that the mechanism for strain release is influenced by both particle size and the degree of delithiation. To achieve stable cycling of LMNO cathodes, it is important to reduce the primary particle size to <1 µm and avoid fully charge the electrode during cycling. Significantly reducing the particle size, however, is not recommended as nano-sized electrode materials often have lower volumetric energy density and enhanced side reactions with the electrolyte. The tradeoffs in energy density, mechanical stability and cycling stability warrant a careful optimization study of the LMNO cathode particles.

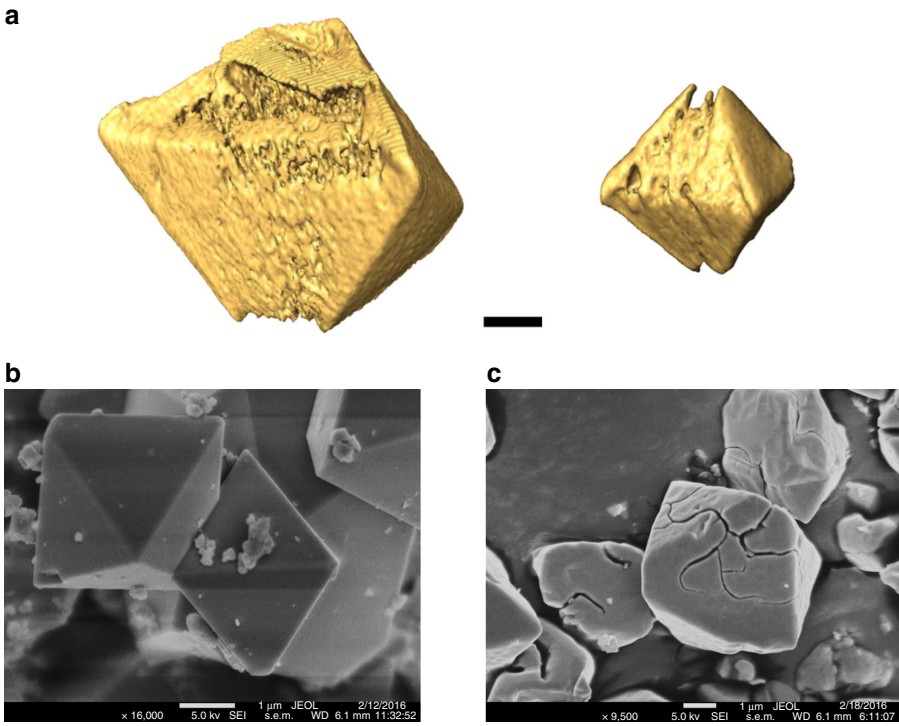

**Figure 6 | Particle fracturing in chemically delithaiated Li$_x$Mn$_{1.5}$Ni$_{0.5}$O$_4$ crystals. (a)** 3D surface renderings of two selected Mn$_{1.5}$Ni$_{0.5}$O$_4$ crystals (the left one at $\sim 4\,\mu m$ and the right one at $2\,\mu m$) show the formation of irregularly shaped cracks on the particle surface, **(b,c)** SEM images of pristine LiMn$_{1.5}$Ni$_{0.5}$O$_4$ and delithiated Mn$_{1.5}$Ni$_{0.5}$O$_4$ crystals, respectively. Scale bars, $1\,\mu m$ **(a–c)**.

## Discussion

Full-field transmission X-ray microscopy technique was successfully used to probe the delithiation pathways and phase transformation mechanism in LMNO microcrystals at nanoscale spatial resolution. Careful analysis of the 2D and 3D chemical phase distribution maps suggested that delithiation initiates at the truncated vertices of the octahedron-shaped microcrystals and then propagates into the bulk of crystal. We propose a concurrent three-phase transformation mechanism as opposed to the well-known shrinking-core or particle-by-particle mechanism. The coexistent of multiple phases on a large MNO crystal results in inhomogeneous strain distribution that was released through the formation of fractures and cracks. The morphological damage is expected to induce mechanical pulverization during battery cycling which may lead to low coulombic efficiency, capacity fade and poor cycle life. Reducing primary particle size and limiting electrode cycling to SOC <1, therefore are important strategies in improving the cycling stability of the LMNO cathode material.

## Methods

**Preparation of crystal samples.** Unless otherwise specified, all chemicals were obtained from Aldrich with a purity of 97% or higher. Well-formed octahedral-shaped LMNO single crystals were prepared according to the procedure described in our previous report[3]. Various levels of chemical delithiation were achieved by reacting the pristine powder with controlled amount of 0.1 M nitronium tetrafluoroborate in acetonitrile solution in an argon-filled glove box ($O_2 < 1$ p.p.m. and $H_2O < 1$ p.p.m.) at room temperature. The resulting reaction mixtures were filtered, thoroughly washed with acetonitrile and then dried overnight in a vacuum oven.

**Materials characterization.** Chemical composition of the samples was determined by an ICP optical emission spectrometer (Perkin-Elmer Optima 5400). X-ray diffraction patterns were collected using a Panalytical X'Pert Pro diffractometer with monochromatized Cu K$\alpha$ radiation. The scans were collected between 30 and 80° ($2\theta$) at a rate of $0.0001° \, s^{-1}$ and a step size of 0.022°.

Hard XAS data on Ni K-edge was collected in transmission mode using a (220) monochromator at SSRL beamline 4–1. Fine powders of as-prepared Li$_x$MNO samples were sandwiched between two Kapton tapes for the measurement. Higher harmonics in the X-ray beam were rejected by detuning the Si (220) monochromator by 35% at the Ni edge. Energy calibration was accomplished by using the first inflection points in the spectra of Ni metal foil reference at 8,333 eV. XANES data were analysed by Sam's Interface for XAS Package or SIXPACK software[49], with the Photoelectron Energy Origin $E_0$ determined by the first inflection point of the absorption edge jump. FF-TXM imaging was performed at the 54 pole wiggler beamline (BL 6-2c) at the SSRL. Detailed beamline configuration can be found in a previous report[50]. Delithiated Li$_x$MNO crystals were dispersed in the cylindrical quartz capillary. Slow and steady He gas flow was applied to the crystals to minimize the radiation-induced heat load on the sample during the scans. The X-ray energy was tuned to Ni K-edge and then focused onto the sample by an elliptically shaped capillary condenser providing illumination for a FOV of $\sim 30 \times 30\,\mu m^2$. 2D transmission images (0.5 s exposure time, 10 repetitions, binning 2, $1{,}024 \times 1{,}024$ pixels) were collected from 8,100 to 8,800 eV in 134 steps for each sample. To remove distortions caused by the flux and beam instabilities, concurrent acquisition of reference images at each energy was also performed through an open area of the sample (outside the capillary) with the same imaging configuration. The repetitions in exposures were carried out to enhance the dynamic range of the existing charge-coupled device and, subsequently, improve the signal to noise ratio in the data. 3D XANES tomography was performed by rotating the quartz capillaries from $-90°$ to 90° with an angular step size of 1° and 58 different energy steps across the Ni K-edge. The tomographic reconstruction and data analysis were performed using TXM-Wizard, an in-house-developed software package. The effective 3D voxel size is $32.5 \times 32.5 \times 32.5\,nm^3$.

**Data availability.** Data supporting the findings of this study are available from the authors on reasonable request. See author contributions for specific data sets.

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

## Acknowledgements

We thank Drs Ryan Davis, D. Van Campen, Johanna Nelson Weker, Jordi Cabana and Young-Sang Yu for the engineering support and helpful discussion on the experiments carried out at beamline 4-1, 6-2c of SSRL. Use of the Stanford Synchrotron Radiation Lightsource, SLAC National Accelerator Laboratory, is supported by the U.S. Department of Energy, Office of Science, Office of Basic Energy Sciences under Contract No. DE-AC02-76SF00515. This work was supported by the Assistant Secretary for Energy Efficiency and Renewable Energy, Office of FreedomCAR and Vehicle Technologies of the U.S. Department of Energy under Contract No. DE-AC02-05CH11231.

## Author contributions

S.K. and G.C. participated in conceiving and designing the experiments. S.K. performed the materials syntheses and characterization, and wrote the paper with the assistance from G.C. and Y.L. S.K., Y.X. and Y.L. designed and performed synchrotron experiments, and analysed the TXM tomography data, under the supervision of Y.L., G.C. supervised the project and the writing of the manuscript. All authors participated in discussions and know the implications of the work.

**Additional information**

**Competing financial interests:** The authors declare no competing financial interests.

