## [Peer Review File · Nature Communications]

Reviewers' comments:

Reviewer #1 (Remarks to the Author):

The manuscript by Kuppan et al reports the investigation of phase transformation mechanism in ordered $\text{LiMn}_{1.5}\text{Ni}_{0.5}\text{O}_4$ at single-particle level during chemical delithiation using 2D and 3D hard X-ray spectroscopic imaging. The manuscript is well-written and the results are quite interesting and insightful for Li-ion cathode materials. I recommend this manuscript for publication in Nature Communications after some revisions as discussed below:

Major comments:

- 1) The authors propose a concurrent 3-phase transformation mechanism for the LNMO cathode during chemical delithiation. While this claim is indeed convincingly supported by the data, I am not sure this will be the (only) phase transformation mechanism that can be observed during electrochemical delithiation (constant-current charge). The phase transformation of LNMO is very likely to be controlled (or at least strongly influenced) by the overpotential, similar to what has been reported for another phase-separating electrode material, LiFePO_4 . The $\text{NO}_2^+/\text{NO}_2$ redox couple has a standard potential of ~ 5.1 V vs Li^+/Li , which is ~ 400 mV higher than the plateau potential (quasi-equilibrium potential) of the LNMO cathode (~ 4.7 V). Therefore, this may be analogous to a phase transformation process under a large overpotential so that concurrent 3-phase mechanism is observed. However, if the overpotential is very small, i.e. electrochemically charging the LNMO cathode at a small C-rate such as 1/50 or 1/100 C, two-phase mechanism (at single-particle level) and particle-by-particle mechanism (at multiple-particle level) may be observed. I believe that the authors need to point out the important role of overpotential in electrochemically-driven phase transformations and discuss the possible difference between chemical and electrochemical delithiation.
- 2) I am still confused about how the standard reference spectrum of $\text{Li}_{0.5}\text{NMO}$ is numerically extrapolated. Did the authors assume a certain crystal structure and site occupancy of $\text{Li}_{0.5}\text{NMO}$ and simulate the XANES spectrum of $\text{Li}_{0.5}\text{NMO}$ or use any other methods? Can the authors provide more details in supporting information?
- 3) The authors mention that "due to the spatial resolution limitation of FFTXM imaging, strain and fracturing survey at the Li_xNMO phase boundaries was not feasible". However, the cracks in Figure 6b seem to be large enough (~ 100 nm?) for FFTXM to see, which has a spatial resolution around 30 nm (in 2D). Can the authors comment on this? How large does the crack need to be if one wants to use FFTXM technique to study crack formation?

Minor comments:

- 1) Page 11, line 236, ref. 19 (Li et al, Nat Commun, 2015, 6, 6883) should not be cited here. It uses hard X-ray spectroscopic imaging to study FeF_3 , not spinel LiMn_2O_4 or LNMO.
- 2) It could be a very interesting future work to use in situ hard X-ray spectroimaging to comparatively study LNMO with different surface facets [(111) vs more (100), like the one shown in ref. 41].

Reviewer #2 (Remarks to the Author):

This paper reports phase transformation mechanism of $4 \mu\text{m}$ -sized $\text{LiNi}_{0.5}\text{Mn}_{1.5}\text{O}_4$ particles using by hard X-ray full-field transition microscopy imaging combined with X-ray absorption near-edge structure (FF-TXM-XANES). The authors demonstrated very nice 2D and 3D chemical phase distribution maps and morphological damages, and proposed a concurrent 3-phase phase transformation mechanism and reducing primary particle size of $1 \mu\text{m}$ to improve cycle life. This work is valuable to be reported to high quality Journal, Nature Communication. However, before it can be accepted, several raised issues should be addressed. There are some ambiguous points in this paper.

1. First of all, it is unclear how much capacity was delivered from the synthesized LNMO and how

about cycle life of the prepared LMNO cathode.

2. In Figure 6, the authors show fractured surface of MNO crystals. How much the MNO was cycled? Just one cycle?

3. In order to discuss mechanical pulverization of the MNO cathode combined with phase transformation, the authors should show 2D and 3D chemical mapping data as function of cycles together with SEM and TEM data of the corresponding MNO samples. Please refer to a recent work (Adv Energy mater. DOI: 10.1002/aenm.201601417) studying degradation mechanism after long-term cycling of layered NMC cathode and discuss mechanical pulverization mechanism.

4. Reviewer doesn't agree the authors' conclusion that reducing primary particle size to below 1 μm is important strategies in improving cycle life because nano-sized electrode (cathode or anode) is significantly decreased volumetric energy density due to lower packing density and cycle life due to parasitic reaction between cathode and electrolyte.

REVIEWERS' COMMENTS:

Reviewer #1 (Remarks to the Author):

I am mostly satisfied with the authors' response to my previous questions. I recommend this manuscript to be published after the authors consider some very minor revisions.

1) The 3D surface renderings in Figure SI-4, which shows cracks on the LNMO particle surface, look very nice and interesting. They could be better shown in together with the SEM images in Figure 6 of the main text.

2) Some revisions need to be done to the paragraph added on page 13-14.

e.g. "The presence of such a large overpotential is likely to promote a high Li flux during the extraction, which may lead to significant variations in local current density and contribute to the particle-level heterogeneity observed in this study."

There is no "current" in chemical delithiation. The delithiation process under a large overpotential is likely limited by bulk diffusion (of Li⁺) (perhaps similar to what has been observed in LiFePO₄).

Therefore, in one single particle, it is possible that the 2nd delithiation process (L0.5NMO to NMO) at the outer surface starts before the 1st delithiation process (LNMO to L0.5NMO) fully finish in the entire particle due to Li-ion transport limitation. This could be one explanation for the particle-level heterogeneity observed in this study.

"It is possible that under small constant-current charging without this voltage driving force (small C-rate)"

It is not correct to say "without this voltage driving force". It should be "under a small overpotential".

The electrochemical reaction (delithiation of LNMO here) will not proceed if without an overpotential.

"in operando techniques such as those recently demonstrated on LiFePO₄" (ref. 30, 31)

Ref. 31 is an ex situ work, not operando (also it is operando not in operando).

Reviewer #2 (Remarks to the Author):

All the comments raised by me have been well addressed. I recommend the paper to be published in Nat Commun.

** See Nature Research's author and referees' website at www.nature.com/authors for information about policies, services and author benefits

Reviewer #1 (Remarks to the Author):

The manuscript by Kuppen et al reports the investigation of phase transformation mechanism in ordered LiMn_{1.5}Ni_{0.5}O₄ at single-particle level during chemical delithiation using 2D and 3D hard X-ray spectroscopic imaging. The manuscript is well-written and the results are quite interesting and insightful for Li-ion cathode materials. I recommend this manuscript for publication in Nature Communications after some revisions as discussed below:

Major comments:

1) The authors propose a concurrent 3-phase transformation mechanism for the LNMO cathode during chemical delithiation. While this claim is indeed convincingly supported by the data, I am not sure this will be the (only) phase transformation mechanism that can be observed during electrochemical delithiation (constant-current charge). The phase transformation of LNMO is very likely to be controlled (or at least strongly influenced) by the overpotential, similar to what has been reported for another phase-separating electrode material, LiFePO₄. The NO₂⁺/NO₂ redox couple has a standard potential of ~5.1 V vs Li⁺/Li, which is ~400 mV higher than the plateau potential (quasi-equilibrium potential) of the LNMO cathode (~4.7 V). Therefore, this may be analogous to a phase transformation process under a large overpotential so that concurrent 3-phase mechanism is observed. However, if the overpotential is very small, i.e. electrochemically charging the LNMO cathode at a small C-rate such as 1/50 or 1/100 C, two-phase mechanism (at single-particle level) and particle-by-particle mechanism (at multiple-particle level) may be observed. I believe that the authors need to point out the important role of overpotential in electrochemically-driven phase transformations and discuss the possible difference between chemical and electrochemical delithiation.

(Authors' response): we thank the reviewer for pointing out the possible mechanistic differences between chemical and electrochemical delithiation and the role of overpotential in phase transformation mechanism. We fully agree that the Li extraction conditions used in this study is similar to electrochemical delithiation under a large overpotential of ~400 mV rather than under a small constant current (slow-rate charging).

A large body of work has already been carried out in order to understand the phase transition mechanism in LiFePO₄, another lithium-ion battery cathode material operating through the first-order transition. The phase transformation mechanism was found to be sensitive to a number of parameters, particularly charging/discharging rate, particle size and temperature. The role of overpotential is indeed important as it is likely to promote a high Li flux during the extraction/insertion, which may lead to significant variations in local current density and contribute to the particle-level heterogeneity observed in this study. It is possible that under small constant-current charging without the voltage driving force, other phase transformation mechanism can dominate or coexist. Systematic studies using different experimental conditions, especially the use of *in operando* techniques such as those recently demonstrated on LiFePO₄ by Li et al [Nat. Mater. 13, 1149-1156 (2014)] and Lim et al [Science 353, 6299, 566-571 (2016)], should provide further information on phase transformation mechanism(s) under electrochemical charge/discharge conditions.

We would like to emphasize that the main advantage of chemical delithiation used in this study is the uniformity in delithiation as all LMNO particles were immersed in the same oxidizing NO₂BF₄ solution. In electrochemical charge/discharge, Li extraction is greatly influenced by

ionic and electronic connectivity which introduces additional variations contributing to the observed delithiation mechanism at the particle-level.

The following paragraph was added on page 13-14 and highlighted in red:

“We would like to point out that the chemical delithiation condition used in this study is similar to that of constant-voltage electrochemical charging under an overpotential. The standard oxidation potential of the $\text{NO}_2^+/\text{NO}_2$ redox couple is ~ 5.1 V vs. Li^+/Li , which is ~ 400 mV higher than that of the plateau potential of the LMNO cathode at ~ 4.7 V. The presence of such a large overpotential is likely to promote a high Li flux during the extraction, which may lead to significant variations in local current density and contribute to the particle-level heterogeneity observed in this study. It is possible that under small constant-current charging without this voltage driving force (small C-rate), other phase transformation mechanism could dominate or coexist. Systematic studies using different experimental conditions, especially the use of *in operando* techniques such as those recently demonstrated on LiFePO_4 ,^{30, 31} should provide further information on phase transformation mechanism(s) under electrochemical charge/discharge conditions. One caution is that in electrochemical charge/discharge, Li extraction is greatly influenced by ionic and electronic connectivity which introduces additional factors controlling the observed delithiation mechanism at the particle-level. Whereas in chemical delithiation, all LMNO particles are immersed in the oxidizing NO_2BF_4 solution with the same concentration and therefore, delithiation uniformity at both particle-level and bulk-level can be achieved.”

2) I am still confused about how the standard reference spectrum of $\text{L}_{0.5}\text{NMO}$ is numerically extrapolated. Did the authors assume a certain crystal structure and site occupancy of $\text{L}_{0.5}\text{NMO}$ and simulate the XANES spectrum of $\text{L}_{0.5}\text{NMO}$ or use any other methods? Can the authors provide more details in supporting information?

(Authors' response): we sincerely apologize for the confusion. The extrapolation of the phase-pure $\text{L}_{0.5}\text{NMO}$ XANES spectrum ($S_{0.5}$) is essentially a mathematical optimization process of obtaining a solution that minimizes the deviation among the results obtained from multiple independent experimental measurements while satisfies the constraints set by a given equation.

More detailed explanation on the method of extracting the XANES spectrum of $\text{L}_{0.5}\text{NMO}$ is included in the revised manuscript. The following sentences were added on page 7-8:

“Specifically, the spectra of phase-pure LMNO and MNO (S_1 and S_0) were acquired directly. All of the experimentally measured spectra of partially delithiated L_xMNO (S_x , $x=0.06, 0.25, 0.51, 0.71, 0.82, 0.9$) were composed of varying contributions from the three primary phases (LMNO, $\text{L}_{0.5}\text{NMO}$ and MNO), with the relative weight fraction of each phase at a given x value (R_x^1 , $R_x^{0.5}$ and R_x^0) obtainable from the Rietveld refinement of the full XRD patterns. This gives us six redundant linear equations that can be expressed as $S_x = (S_0 \times R_x^0) + (S_{0.5} \times R_x^{0.5}) + (S_1 \times R_x^1)$. The only unknown parameter in these equations is $S_{0.5}$, which is the signature spectrum of the phase-pure $\text{L}_{0.5}\text{NMO}$. $S_{0.5}$ was then solved by minimizing the deviation among the results derived from multiple independent experimental measurements while satisfying the constraints set by the equation described above. The calculation was implemented using Matlab's optimization tool box.”

3) The authors mention that “due to the spatial resolution limitation of FFTXM imaging, strain and fracturing survey at the Li_xMNO phase boundaries was not feasible”. However, the cracks in Figure 6b seem to be large enough (~ 100 nm?) for FFTXM to see, which has a spatial resolution around 30 nm (in 2D). Can the authors comment on this? How large does the crack need to be if one wants to use FFTXM technique to study crack formation?

(Authors’ response): we greatly appreciate the reviewer for catching this misleading statement in the paper.

It is not feasible to study the strain in the particle using FF-TXM because the contrast mechanism of this technique is not sensitive to the local crystal structure distortion. The FF-TXM detects the difference in the X-ray absorption and, by extension, the absorption coefficient’s variation as a function of the X-ray energy, which is sensitive to the local oxidation state. For the survey of the crack in the particle, FF-TXM can indeed provide useful insight in 3D. For most part of our study, we operated the FF-TXM around the Ni K-edge for the chemical sensitivity. However, we can optimize the microscope at lower X-ray energy in order to study particle morphology, such as the formation of cracks. This is because better absorption contrast and higher spatial resolution could be achieved at lower energy. Below we show the three-dimensional surface rendering of two selected MNO particles (one at about 4 microns and the other at about 2 microns) which nicely complement the observation in our SEM studies, as the formation of cracks is also clearly observed on the surface.

Figure SI-4. 3D surface renderings of two selected MNO crystals prepared by chemical delithiation (the left one at about 4 μm and the right one at 2 μm) show the formation of irregularly shaped cracks on the particle surface.

We have modified the paragraphs to incorporate these changes. The following sentences were included on page 14-15, along with the changes made in the reference section and the addition of Figure SI-4 in the Supporting Information.

“A recent study using the *in situ* coherent X-ray diffractive imaging (CXDI) technique showed that the strain energy is relatively low at the early stages of LMNO delithiation but it increases more than ten-fold during the topotactic phase transformation process.^{16, 47} Striking inhomogeneity in the strain distribution throughout the particle was also reported. Further evidence was provided by *in situ* multi-beam optical stress sensor (MOSS) on LMNO thin-film electrodes, which showed that the stress in LMNO increases with lithium removal and it can reach up to 126 MPa at fully delithiated state (MNO).⁴⁸ Strain survey on Li_xMNO crystals was not feasible with FF-TXM as the contrast mechanism of the technique is not sensitive to the local crystal structure distortion. However, morphology survey on the particles, such as the formation

of cracks, can be carried out especially if the microscope is operated at lower X-ray energy than that of the Ni K-edge. This is because better absorption contrast and higher spatial resolution are achievable at lower energy. Figure SI-4 shows the three-dimensional surface rendering of two selected fully delithiated MNO crystals (4 and 2 μm in size) prepared by chemical oxidation. The formation of irregularly shaped cracks on the particle surface is clearly shown.

Further evidence on delithiation-induced morphological changes was provided by SEM studies, as shown by the images obtained on pristine LMNO and chemically delithiated MNO in Figure 6.”

The following references were modified in the reference section:

47. Ulvestad A, *et al.* In situ strain evolution during a disconnection event in a battery nanoparticle. *Phys. Chem. Chem. Phys.* **17**, 10551-10555 (2015).
48. Li J, Zhang Q, Xiao X, Cheng Y-T, Liang C, Dudney NJ. Unravelling the Impact of Reaction Paths on Mechanical Degradation of Intercalation Cathodes for Lithium-Ion Batteries. *J. Am. Chem. Soc.* **137**, 13732-13735 (2015).

Minor comments:

1) Page 11, line 236, ref. 19 (Li *et al*, *Nat Commun*, 2015, 6, 6883) should not be cited here. It uses hard X-ray spectroscopic imaging to study FeF₃, not spinel LiMn₂O₄ or LNMO.

(Authors' response): we thank the reviewer for catching this mistake. The reference has been removed in the revised manuscript.

2) It could be a very interesting future work to use in situ hard X-ray spectroimaging to comparatively study LNMO with different surface facets [(111) vs more (100), like the one shown in ref. 41].

(Authors' response): we agree with the reviewer's suggestion and we plan to look into this comparative study in the near future.

Reviewer #2 (Remarks to the Author):

This paper reports phase transformation mechanism of 4 μm -sized LiNi_{0.5}Mn_{1.5}O₄ particles using by hard X-ray full-field transition microscopy imaging combined with X-ray absorption near-edge structure (FF-TXM-XANES). The authors demonstrated very nice 2D and 3D chemical phase distribution maps and morphological damages, and proposed a concurrent 3-phase phase transformation mechanism and reducing primary particle size of 1 μm to improve cycle life. This work is valuable to be reported to high quality Journal, Nature Communication. However, before it can be accepted, several raised issues should be addressed. There are some ambiguous points in this paper.

1. First of all, it is unclear how much capacity was delivered from the synthesized LMNO and how about cycle life of the prepared LMNO cathode.

(Authors' response): we thank the reviewer for catching this omission. The following sentence was added on page 6 along the addition of a new reference on page 23:

“Pristine LMNO crystals with a cubic spinel phase and structurally ordered in a $P4_332$ space group were synthesized by the molten-salt method.³ Previous electrochemical studies showed that the sample delivered ~140 mAh/g at C/5 rate and experienced an average per-cycle loss of ~0.05% and a coulombic efficiency of 99% in the first 170 cycles.³⁸”

38. Kupan S., Duncan H. & Chen G., Controlling Side Reactions and Self-discharge in High-voltage Spinel Cathodes: Critical Role of Surface Crystallographic Facets. *Phys. Chem. Chem. Phys.* **17**, 26471 (2015).

2. In Figure 6, the authors show fractured surface of MNO crystals. How much the MNO was cycled? Just one cycle?

(Authors' response): the SEM image shown in Figure 6b was taken from the crystals recovered after first chemical delithiation. The sample was not subjected to electrochemical cycling. We apologize for this confusion.

To clarify, we have modified the text on page 15 and the figure caption on page 30:

“Further evidence on delithiation-induced morphological changes was provided by SEM studies, as shown by the images obtained on pristine LMNO and chemically delithiated MNO in Figure 6.”

Figure 6. SEM images of: a) pristine LMNO and b) chemically delithiated MNO crystals.

3. In order to discuss mechanical pulverization of the MNO cathode combined with phase transformation, the authors should show 2D and 3D chemical mapping data as function of cycles together with SEM and TEM data of the corresponding MNO samples. Please refer to a recent work (*Adv Energy mater.* DOI: 10.1002/aenm.201601417) studying degradation mechanism after long-term cycling of layered NMC cathode and discuss mechanical pulverization mechanism.

(Authors' response): we agree that in order to determine cycling-induced mechanical pulverization of LMNO cathode combined with phase transformation, it is necessary to collect 2D and 3D chemical mapping data as a function of cycle number together with SEM and TEM images of the corresponding samples. In this study, our focus is on particle-level phase transformation mechanism and morphological changes during the first delithiation. Understanding the long-term cycling related changes is obviously important for the performance of LMNO cathode and we plan to look into this in our future studies. We thank the reviewer for the comment.

4. Reviewer doesn't agree the authors' conclusion that reducing primary particle size to below 1 μm is important strategies in improving cycle life because nano-sized electrode (cathode or anode) is significantly decreased volumetric energy density due to lower packing density and cycle life due to parasitic reaction between cathode and electrolyte.

(Authors' response): We agree with the reviewer that nano-sized electrode materials have disadvantages in terms of lower volumetric energy density and enhanced side reactions with the electrolyte. We believe that reducing LMNO particle size to some extent, below 1 μm but above the nano-regime, can benefit the cycling stability of LMNO. The tradeoffs in energy density, mechanical stability and cycling stability, however, warrant a careful optimization study.

The following sentences were added on page 15-16 to reflect this thought process:

“Significantly reducing the particle size, however, is not recommended as nano-sized electrode materials often have lower volumetric energy density and enhanced side reactions with the electrolyte. The tradeoffs in energy density, mechanical stability and cycling stability warrant a careful optimization study of the LMNO cathode particles.”

On page 16, we also modified the following sentence to avoid the confusion:

“Reducing primary particle size and limiting electrode cycling to SOC below 1, therefore are important strategies in improving the cycling stability of the LMNO cathode material.”

Additional changes made in the revise manuscript:

- 1) Page 1: we have added the following secondary affiliation of Ms. Yahong Xu which was approved by all authors.
^c College of Mechanical Engineering, Donghua University, Shanghai 200051, China
- 2) Page 5: we have removed reference 22 because it is not related to LFP.
- 3) Page 5 and 23: To reflect the most current application status of the FF-TXM technique, we have added “and $\text{Li}_2\text{Ru}_{0.5}\text{Mn}_{0.5}\text{O}_3$.³⁷” on page 5 and cited an additional reference on page 23:
37. Xu et al., Nano Energy 28, 164-171 (2016).
- 4) Page 5 and 22: To emphasize the active development of scientific big data mining associated with spectro-microscopy, we have added the following additional reference on the topic:
27. Duan et al., Sci. Rep. 6, 34406, (2016).

Reviewer #1 (Remarks to the Author):

I am mostly satisfied with the authors' response to my previous questions. I recommend this manuscript to be published after the authors consider some very minor revisions.

1) The 3D surface renderings in Figure SI-4, which shows cracks on the LNMO particle surface, look very nice and interesting. They could be better shown in together with the SEM images in Figure 6 of the main text.

(Authors' response): we thank the reviewer for the great suggestion. We have moved Figure SI-4 from the supporting information to the main text and combined it with the SEM images to create the new Figure 6. We modified the text on p15 and Figure 6 caption to the following:

“Figure 6a shows the three-dimensional surface rendering of two selected fully delithiated MNO crystals (4 and 2 μm in size) prepared by chemical oxidation. The formation of irregularly shaped cracks on the particle surface is clearly shown. Further evidence on delithiation-induced morphological changes was provided by SEM studies, as shown by the images obtained on pristine LMNO and chemically delithiated MNO in Figure 6b and 6c, respectively.”

“Figure 6. a) 3D surface renderings of two selected MNO crystals prepared by chemical delithiation (the left one at about 4 μm and the right one at 2 μm) show the formation of irregularly shaped cracks on the particle surface, b) and c) SEM images of pristine LMNO and chemically delithiated MNO crystals, respectively.”

2) Some revisions need to be done to the paragraph added on page 13-14. e.g. “The presence of such a large overpotential is likely to promote a high Li flux during the extraction, which may lead to significant variations in local current density and contribute to the particle-level heterogeneity observed in this study.” There is no “current” in chemical delithiation. The delithiation process under a large overpotential is likely limited by bulk diffusion (of Li^+) (perhaps similar to what has been observed in LiFePO_4). Therefore, in one single particle, it is possible that the 2nd delithiation process ($\text{L}0.5\text{NMO}$ to NMO) at the outer surface starts before the 1st delithiation process (LNMO to $\text{L}0.5\text{NMO}$) fully finish in the entire particle due to Li-ion transport limitation. This could be one explanation for the particle-level heterogeneity observed in this study. “It is possible that under small constant-current charging without this voltage driving force (small C-rate)” It is not correct to say “without this voltage driving force”. It should be “under a small overpotential”. The electrochemical reaction (delithiation of LNMO here) will not proceed if without an overpotential. ”in operando techniques such as those recently demonstrated on LiFePO_4 ”(ref. 30, 31). Ref. 31 is an ex situ work, not operando (also it is operando not in operando).

(Authors' response): we thank the reviewer for catching these mistakes and the knowledgeable addition. We have addressed each one of them and modified the wording in the revision.

- 1) We have removed “local current” and added “local concentration and contribute to the particle-level heterogeneity observed in this study” in the revised text.
- 2) We have removed “without this voltage driving force” and added “with only a small overpotential”.
- 3) We have removed reference 31 and changed “in operando” to “operando”.

Below is the modified paragraph on p13-14:

“We would like to point out that the chemical delithiation condition used in this study is similar to that of constant-voltage electrochemical charging under an overpotential. The standard oxidation potential of the $\text{NO}_2^+/\text{NO}_2$ redox couple is ~ 5.1 V vs. Li^+/Li , which is ~ 400 mV higher than that of the plateau potential of the LMNO cathode at ~ 4.7 V. The presence of such a large overpotential is likely to promote a high Li flux during the extraction, which may lead to significant variations in local concentration and contribute to the particle-level heterogeneity observed in this study. Furthermore, the delithiation process under these conditions is likely limited by the bulk diffusion of Li^+ which may lead to the initiation of the 2nd phase transformation ($\text{L}_{0.5}\text{MNO}$ to MNO) at the outer surface before the 1st phase transformation (LMNO to $\text{L}_{0.5}\text{MNO}$) in the entire particle can complete. This further contributes to the particle-level heterogeneity observed in this study. It is possible that under low constant-current charging (small C-rate) with only a small overpotential, other phase transformation mechanism could dominate or coexist. Systematic studies using different experimental conditions, especially the use of *operando* techniques such as those recently demonstrated on LiFePO_4 ³⁰ should provide further information on phase transformation mechanism(s) under electrochemical charge/discharge conditions. One caution is that in electrochemical charge/discharge, Li extraction is greatly influenced by ionic and electronic connectivity which introduces additional variations contributing to the observed delithiation mechanism at the particle-level. Whereas in chemical delithiation, all LMNO particles are immersed in the oxidizing NO_2BF_4 solution with the same concentration and therefore, delithiation uniformity at both particle-level and bulk-level can be achieved.”

Reviewer #2 (Remarks to the Author):

All the comments raised by me have been well addressed. I recommend the paper to be published in Nat Commun.